# How Useful are Gradients for OOD Detection Really?

## Abstract

One critical challenge in deploying machine learning models in real-life applications is out of distribution (OOD) detection. Given a predictive model which is accurate on in distribution (ID) data, an OOD detection system can further equip the model with the option to defer prediction when the input is novel and the model has low confidence. Notably, there has been some recent interest in utilizing gradient information in pre-trained models for OOD detection. While these methods are competitive, we argue that previous works conflate their performance with the necessity of gradients. In this work, we provide an in-depth analysis and comparison of gradient based methods and elucidate the key components that warrant their OOD detection performance. We further demonstrate that a general, non-gradient-based family of OOD detection methods are just as competitive, casting doubt on the usefulness of gradients for OOD detection.

## 1 Introduction

Recent advances in algorithms, models, and training infrastructure have brought about unprecedented performance of machine learning (ML) methods, across a wide range of data types and tasks. Despite their demonstrated potential on benchmark settings and domains, one obstacle which limits ML methods' applicability in real-world applications is the *uncertainty* or *confidence* of the predictions. Without any deliberate mechanisms, ML models will output a prediction for any given input, and the question of whether this prediction can be *trusted* will be especially critical in many high-risk decision-making settings (e.g. self-driving cars (Agarwal et al., 2021), physical sciences (Char et al., 2021; Boyer et al., 2021), and healthcare (Zhou et al., 2020)). This risk is further exacerbated with deep learning where the interpretability of models are often limited (Rudin, 2019)).

It is unrealistic for one to expect to train a model that has perfect predictions for all possible inputs, partly because real-world datasets are limited in their scope. Thus in lieu of trying to make predictions for all test inputs, one can attempt to first detect whether the input is covered by the support of the training data. This is the motivation behind *OOD detection*. Among the diverse approaches to OOD detection for image recognition, a recent line of work has suggested utilizing the information in gradients to derive efficient and performant methods for OOD detection (Liang et al., 2017; Lee & AlRegib, 2020; Agarwal et al., 2020; Lee & AlRegib, 2021; Huang et al., 2021; Sun et al., 2022; Kokilepersaud et al., 2022).

We motivate our work by first exploring the claim that gradients are useful for OOD detection. Through a comparison with various extensions of gradient-based scores, we analyze the key components that actually drive the performance of these methods, and we argue that gradient computations are not essential in deriving performant post hoc OOD detectors. Rather, these methods ultimately rely on the magnitude of the learned feature embedding and the predicted output distribution. *We thereby refute many of the intuitions that previous works motivate their methods with*. Based on our analysis, we advocate for the study of a more general, non-gradient-based framework for producing performant score functions and provide a comprehensive empirical evaluation of various instantiations of the score within this framework.

The rest of this paper is structured as follows. We first provide a formal statement of the problem setting and the related works in Section 2. We then introduce existing and new gradient-based detectors and discuss how both can be simplified into intuitive forms (Section 3). Following this, we perform empirical evaluations of the methods (Section 4) and discuss their implications in Section 5.

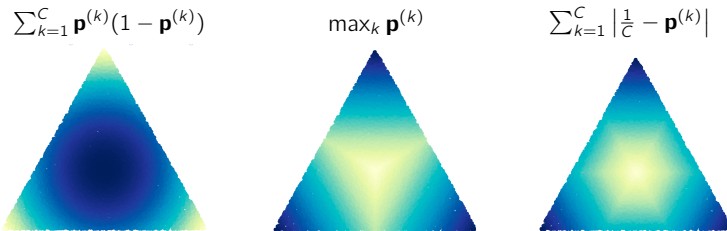

Figure 1: **Output Score Components** The probability output components ($V$ components from Section 4.2) are shown on a 3-class probability simplex where each of the 3 vertices signifies probability 1 to a single class. From left to right, these are the output score terms used for EXGRAD, MSP, and GRADNORM. Lighter to darker shades indicate lower to higher values.

## 2    PRELIMINARIES AND RELATED WORK

**Problem Setting and Notation** We focus on the classification setting where the task is to predict a class label $y \in [C]$ given an input $\mathbf{x} \in \mathbb{R}^d$, where $C$ is the total number of possible classes and $d$ is the dimensionality of the input. We assume we have access to training data $\{(\mathbf{x}_i, y_i)\}_{i=1}^N$ in order to train the parameters $\theta$ of a deep neural network $f_\theta : \mathbb{R}^d \to \mathbb{R}^C$. Here, $\theta$ is a collection of all of the weights from each layer in the network, thus, $\theta = \{W_l : l \in [L]\}$, where $l$ denotes the index of each layer, and we assume the network has $L$ layers in total.

Given an input $\mathbf{x}$, the network predicts a probability vector, $\mathbf{p}$, via the softmax function of the network outputs: $\mathbf{p}^{(k)}(\mathbf{x}) = \mathbb{P}(Y = k | \mathbf{x}) = \frac{\exp\left(f_\theta^{(k)}(\mathbf{x})/T\right)}{\sum_{k'=1}^C \exp\left(f_\theta^{(k')}(\mathbf{x})/T\right)}$ , where the superscripts denote the index of the vector and $T$ is the temperature. If not otherwise specified, we will assume $T = 1$. Although $\mathbf{p}(\mathbf{x})$ depends on both $\theta$ and $\mathbf{x}$ , we will always exclude the former and often exclude the latter when it is clear from context or unimportant. Lastly, we often abuse notation and use $Y \sim \mathbf{p}(\mathbf{x})$ to mean $Y$ is sampled from the categorical distribution parameterized by $\mathbf{p}(\mathbf{x})$.

**The OOD Detection Problem** In a real-world scenario, the model may be given an input, $\tilde{\mathbf{x}} \in \mathbb{R}^d$, during deployment that is substantially different from any of the datapoints in the training set. For example, a classifier trained to identify numeric digits may be given an image of a cat. Since the model's prediction $\mathbf{p}(\tilde{\mathbf{x}})$ cannot be trusted, it would be advantageous to flag such instances and defer prediction. This is the problem that OOD detection addresses.

More formally, the goal in OOD detection is to derive a binary classifier which labels whether a given input, $\tilde{\mathbf{x}}$, is ID (in distribution) or OOD. This goal is commonly addressed by learning a score function $\mathcal{S} : \mathbb{R}^d \to \mathbb{R}$ which quantifies the degree to which the input is OOD. Existing works have approached this problem of learning the mapping $\mathcal{S}$ from various perspectives, which generally vary by *where* the signal to generate $\mathcal{S}$ is extracted from. Here, we introduce broad groupings of methodologies to provide context for our work, and we refer the reader to Yang et al. (2021); Salehi et al. (2021) for an in-depth survey of the field.

One class of methods focuses on the input space and learns score functions based on characteristics that can be derived from the input features. For example, distance-based methods are based on the intuition that OOD data should lie "far away" from ID data and define $\mathcal{S}$ with distances between the input point and reference points that are representative of ID (Lee et al., 2018; Techapanurak et al., 2020; Van Amersfoort et al., 2020). Meanwhile, density-based methods utilize probabilistic models to describe the density of ID data and argue that OOD points should occur in areas of low densities. Thus generative models are often used and the score function is often derived with the predicted likelihood of the test input point (Ren et al., 2019; Serrà et al., 2019; Zisselman & Tamar, 2020).

Another class of methods aims to directly influence a predictive model's behavior to OOD data through explicit training. These methods often assume access to OOD examples during training and incorporates them into a predictive model's training procedure to maximize the separability between ID and OOD inputs. This is usually achieved by setting aside actual samples from an OOD test distribution (Hendrycks et al., 2018; Yu & Aizawa, 2019; Liu et al., 2020), or if they are not available, by synthesizing OOD examples via adversarial training, perturbations, or sampling from boundaries or low density regions (Lakshminarayanan et al., 2016; Lee et al., 2017; Vernekar et al., 2019).

Many works have instead turned attention to the information that can be extracted from a predictive model that is fully trained on ID data. With the intuition that an ideal predicted distribution should have low uncertainty (heavily concentrated on the predicted class) for an ID point and have high uncertainty (close to a uniform distribution) for an OOD input, some methods focus on generating scores with the predicted distribution of pre-trained models Hendrycks & Gimpel (2016); Linmans et al. (2020); Liu et al. (2020). Meanwhile, some works have examined the information available when backpropagating gradients of a loss function. Liang et al. (2017); Agarwal et al. (2020) utilize information in the gradient w.r.t. the input, while Lee & AlRegib (2020); Huang et al. (2021) examine gradients w.r.t. the model parameters. Huang et al. (2021), in particular, proposed GRADNORM, an efficient post hoc score function which simply measures the norm of model parameter gradients and thus requires no additional training or hyperparameter tuning. Despite this, GRADNORM achieves State-of-the-Art (SotA) performance among post hoc methods. We explain GRADNORM in more detail in the next section.

# 3 GRADIENT BASED OOD METHODS

Lee & AlRegib (2020) initially proposed using model gradients as a signal to detect OOD data. Specifically, they provide a fully trained network with an input point and backpropagate the cross entropy loss between a uniform probability distribution and the network's predicted class probabilities. We summarize their intuition for this algorithm in the following hypothesis:

**Hypothesis 3.1.** ***The Feature-Extraction Hypothesis*** *If learning a point $(\tilde{\mathbf{x}}, \tilde{y})$ requires a large change in a well-trained network, then $\tilde{\mathbf{x}}$ must have been novel, because most of a deep network is dedicated to feature extraction, which by assumption should already perform well on ID data.*

Meanwhile, GRADNORM contends the *opposite* case: they argue that trying to fit the predicted distribution of *ID* data to a uniform distribution label will necessitate higher gradients to the model parameters than with OOD data. We note that this intuition relies on the predicted distributions actually displaying high entropy for OOD inputs and low entropy for ID inputs.

Despite conflicting intuitions, both works ultimately propose taking the KL divergence (identically, the cross-entropy loss) w.r.t. the uniform distribution and measuring a norm of the model gradients as a score function. Through extensive empirical ablations, GRADNORM hones in on the $L_1$ norm of the last layer gradients of the KL-divergence w.r.t. the uniform distribution as the score function,

$$\mathcal{S}_{GN}(\mathbf{x}) = \left\| \frac{\partial D_{KL}(\mathbf{u} \| \text{softmax}(f_\theta(\mathbf{x})))}{\partial W_L} \right\|_1 . \tag{1}$$

We can expand the divergence term and simplify this score into the following expression:

$$\mathcal{S}_{GN}(\mathbf{x}) = \left\| \mathbb{E}_{Y \sim \text{unif}} \left[ \nabla_{W_L} \log \mathbf{p}^{(Y)} \right] \right\|_1 . \tag{2}$$

That is, GRADNORM measures the norm of the expected gradient according to a uniform labeling distribution. Note that $\nabla_{W_L}$ denotes the gradient w.r.t. the parameters in the final layer of $f_\theta$.

**Another Gradient Approach** We note that Eq.2 is not a unique measure that leverages gradients. To aid our analysis, we also consider the following variant:

$$\mathcal{S}_{EG}(\mathbf{x}) = \mathbb{E}_{Y \sim \mathbf{p}(\mathbf{x})} \left[ \left\| \nabla_\theta \log \mathbf{p}^{(Y)} \right\|_p \right] . \tag{3}$$

We will show later that this score also has interpretable properties, and helps us understand the central components of gradient-based score functions. Due to the outer positioning of the expectation, we refer to the score function in Eq. 3 as EXGRAD. Two key differences in this score are 1) the label distribution of $Y$ comes from the model's own predicted distribution, $\mathbf{p}$, not the uniform distribution, and 2) this calculates the expected norm of the gradient, whereas Eq. 2 calculates the norm of the expected gradient. Other gradient-based scores are readily plausible, and we visit a suite of scores with empirical comparisons in Section 4.1.

## 3.1 DECOMPOSITION OF GRADIENT METHODS

To give further grounds for their method, Huang et al. (2021) analyze Eq. 2 and show that it can be decomposed into two terms: one that is characterized by the size of the magnitude of the encoding

fed to the last layer of the network, and one that is characterized by the output of the network. In particular, they derive

$$\mathcal{S}_{GN}(\mathbf{x}) = \frac{1}{T} \|\mathbf{h}\|_1 \sum_{k=1}^{C} \left| \frac{1}{C} - \mathbf{p}^{(k)} \right| \tag{4}$$

where $\mathbf{h}$ is the encoding being fed to the last layer of the network. Following their notation, we will denote $U$ to be the part of the score characterized by $\mathbf{h}$ and $V$ to be the part characterized by the network output, i.e. here $U = \|\mathbf{h}\|_1$, $V = \frac{1}{2} \sum_{k=1}^{C} \left| \frac{1}{C} - \mathbf{p}^{(k)} \right|$, and $\mathcal{S}_{GN} = \frac{2}{T} UV$. Huang et al. (2021) show that $U$ and $V$ have the same trend where their values are large for ID points and lower for OOD points. While each can be used as OOD detectors alone, the product of the two terms results in stronger performance.

Although not mentioned in their analysis, the $V$ term in GRADNORM's score is simply the total variation (TV) distance between a discrete uniform distribution and the model's predicted distribution. This characterization of $V$ gives insight as to why GRADNORM works. When the input image is in distribution, the network will likely have higher confidence, making the TV distance between $\mathbf{p}$ and the discrete uniform large.

**Decomposition of EXGRAD** Doing a similar analysis, our alternative gradient-based detector, EXGRAD, can be broken down in a similar way. In particular,

$$\mathcal{S}_{EG}(\mathbf{x}) = \frac{2}{T} UV \qquad U = \|\mathbf{h}\|_1 \qquad V = \sum_{k=1}^{C} \mathbf{p}^{(k)}(1 - \mathbf{p}^{(k)}) \tag{5}$$

The full derivation is shown in Appendix B. Like with GRADNORM, the $V$ term for EXGRAD turns out to be an interpretable quantity. Let $B_k \sim \text{Bernoulli}(\mathbf{p}^{(k)})$ be the random variable corresponding to the event that $\mathbf{x}$ belongs to class $k$. Then, $V = \sum_{k=1}^{C} \text{Var}(B_k)$. Intuitively, when the input image is in distribution and there is high confidence on a single class, the variance of each Bernoulli random variable will be low. Note that this is the opposite trend of the TV distance and $\|\mathbf{h}\|_1$. A visual comparison of these scores can be seen in Figure 1.

This general $UV$-style score will be of particular interest to us throughout this paper, and we will often refer to it as an "Encoding-Output" composition as it relies on both the networks encoding for the image (at least at the penultimate layer) and its output. We note that recent work on the "Familiarity Hypothesis" in Dietterich & Guyer (2022), as well as work on the role of the feature norm in open-set recognition and OOD detection in Vaze et al. (2022) further motivates the study of $U$ and suggests a plausible alternative theory to the gradient-based "Feature-Extraction Hypothesis".

## 4 EMPIRICAL EVALUATION

The previous section introduced variations to existing gradient-based scores and decomposed the scores into interpretable components. This naturally begs the questions, which scores actually perform well, and what should their favorable performance be attributed to? In this section, we explore these questions by comparing the performance of multiple gradient-based scores against simpler, non-gradient based approaches on OOD detection tasks in image classification. Specifically, the goals of these experiments are as follows:

1. to investigate whether different gradient-based scores are useful for OOD detection or whether only particular variants perform well (i.e. is it essential that the gradient-based detector is derived by taking the derivative of the KL divergence?);

2. to examine the plausibility of the feature-extraction hypothesis in explaining the strong performance of gradient-based OOD detection methods;

3. to investigate the claim that gradient-based approaches offer unique performance advantages in the context of post hoc OOD detection tasks.

Indeed, we show that SotA OOD detection performance is achievable by leveraging information solely from the predicted class distribution and latent encoding, calling into question the additional utility gained from gradient based score functions. Furthermore, we find that there is significant variability in performance across tasks for different gradient-based score functions. Moreover, we

show that gradient-based methods are often worse than computationally simpler approaches that require no backpropagation. Lastly, we perform experiments that challenge the plausibility of one emerging theory that attempts to explain the role of gradients in SotA OOD methods.

## 4.1 INVESTIGATING THE PERFORMANCE OF GRADIENT-BASED SCORE VARIANTS

**Experimental Set Up** A key motivation of our work is in the design of post hoc OOD detection mechanisms. In light of this, for each OOD method, we take a deep neural network pre-trained on a particular dataset (the ID dataset) and use data from one of the remaining datasets as OOD data. These models were obtained from a popular open-source library [1] of pre-trained neural network image classifier models and achieve competitive performance on ID data.

We investigate the performance of seven gradient-based OOD scores, one of which has been previously described in the literature (GRADNORM) and another which we described in detail in Section 3 (EXGRAD). We include additional experimental results on natural variants that can be derived by simple design choices in implementation, specifically by interchanging norms and expectations, choice of norm and choice of distribution to generate the synthetic label at test time. Finally, we compare these gradient-based methods against two non-gradient based approaches inspired by Huang et al. (2021) and our analysis in Section 3. We refer to these scores as "$V$ term" scores.

We used MNIST Deng (2012), SVHN Netzer et al. (2011) and CIFAR-10 Krizhevsky (2009) as our base datasets for the first experimental setup, and use 10,000 samples from the test split of each dataset to form our ID and OOD datasets. Each OOD method defines a score function $\mathcal{S}(\tilde{\mathbf{x}})$ which we then use to define an OOD classifier $\mathbf{I}[\mathcal{S}(\tilde{\mathbf{x}}) > \epsilon]$. We then calculate AUROCs for each method, varying the threshold $\epsilon$. Table 1 describes the mean and standard deviation of the AUROC for each method across the six ID-OOD dataset combinations. Full experimental results showing performance for each ID-OOD combination are available in Appendix A. The columns "Deep" and "Shallow" refer to variants of each method that perform gradients w.r.t. all parameters of the network (Deep) or w.r.t. just the parameters of the final layer (Shallow), i.e. the weights of the layer that generate logits.

**Analysis of Results** We first observe that, on average over the 6 ID-OOD benchmark tasks, EXGRAD outperforms the previous SotA gradient-based post hoc OOD method, GRADNORM. Moreover, *no method consistently dominates all other methods across ID-OOD splits* (see Appendix A). This suggests that the singular focus in the literature on backpropagating KL divergence losses is unwarranted.

Our next observation is that, in instances where extending gradients to the entire network produces change in performance, such changes are at best modest gains, and can in fact *hurt* performance. This aligns with results found in Huang et al. (2021). Given that the last layer of a deep neural network is merely taking linear combinations of learned features, this observation serves as further evidence against the Feature-Extraction Hypothesis.

Lastly, we note that the highest performing score functions are the *non-gradient-based* approaches. In particular, the two score functions inspired from the decomposition analyses improve beyond their best gradient-based variants by 2.9 and 0.7 percentage points respectively.

These observations provide the basis for a general framework to achieve high-performing post hoc OOD detectors that do not rely on calculating test-time gradients: combining a norm of learned encodings with a function of the predicted class distribution. In the following section, we provide additional experimental details that leverage this proposed template to further illustrate the ability to achieve high performance without relying on test-time gradients.

## 4.2 EXPLORING ENCODING-OUTPUT COMPOSITIONS

**The ImageNet Benchmark** In addition to the previous experimental set up, in this section we also evaluate on the large-scale ImageNet benchmark proposed by Huang & Li (2021). For this benchmark, the ImageNet-1k dataset (Deng et al., 2009) is used for the ID dataset. This ID dataset is different from MNIST, CIFAR10, and SVHN in that it is composed of higher resolution images and has $C = 1,000$ classes instead of $C = 10$. The iNaturalist (Van Horn et al., 2017), SUN (Xiao et al., 2010), Places (Zhou et al., 2017), and Textures (Cimpoi et al., 2014) datasets are used as OOD datasets. This benchmark uses the 50,000 images in the ImageNet validation set as the ID data and

---

[1]`https://github.com/aaron-xichen/pytorch-playground`

| Score Expression | AUROC | Gradient Depth |
|---|---|---|
| $\|\mathbb{E}_{Y\sim\mathbf{p}}\left[\nabla_\theta\log\mathbf{p}^{(Y)}\right]\|_2^2$ | 0.725 ($\pm$ 0.086) | Deep |
| $\|\mathbb{E}_{Y\sim\mathbf{p}}\left[\nabla_\theta\log\mathbf{p}^{(Y)}\right]\|_2^2$ | 0.741 ($\pm$ 0.081) | Shallow |
| $\mathbb{E}_{Y\sim\text{Uniform}}\left[\|\nabla_\theta\log\mathbf{p}^{(Y)}\|_1\right]$ | 0.825 ($\pm$ 0.120) | Deep |
| $\mathbb{E}_{Y\sim\text{Uniform}}\left[\|\nabla_\theta\log\mathbf{p}^{(Y)}\|_1\right]$ | 0.850 ($\pm$ 0.135) | Shallow |
| $\mathbb{E}_{Y\sim\text{Uniform}}\left[\|\nabla_\theta\log\mathbf{p}^{(Y)}\|_2^2\right]$ | 0.867 ($\pm$ 0.148) | Deep |
| $\mathbb{E}_{Y\sim\text{Uniform}}\left[\|\nabla_\theta\log\mathbf{p}^{(Y)}\|_2^2\right]$ | 0.887 ($\pm$ 0.107) | Shallow |
| $\|\nabla_\theta\mathbb{E}_{Y\sim\text{Uniform}}\left[\log\mathbf{p}^{(Y)}\right]\|_1$  (GRADNORM) | 0.892 ($\pm$ 0.087) | Deep |
| $\|\nabla_\theta\mathbb{E}_{Y\sim\text{Uniform}}\left[\log\mathbf{p}^{(Y)}\right]\|_1$  (GRADNORM) | 0.906 ($\pm$ 0.092) | Shallow |
| $\mathbb{E}_{Y\sim\mathbf{p}}\left[\frac{\log\mathbf{p}^{(Y)}}{\mathbf{p}^{(Y)}}\|\nabla_\theta\log\mathbf{p}^{(Y)}\|_2^2\right]$ | 0.910 ($\pm$ 0.090) | Shallow |
| $\mathbb{E}_{Y\sim\mathbf{p}}\left[\|\nabla_\theta\log\mathbf{p}^{(Y)}\|_2^2\right]$ | 0.919 ($\pm$ 0.041) | Shallow |
| $\mathbb{E}_{Y\sim\mathbf{p}}\left[\|\nabla_\theta\log\mathbf{p}^{(Y)}\|_2^2\right]$ | 0.921 ($\pm$ 0.034) | Deep |
| $\mathbb{E}_{Y\sim\mathbf{p}}\left[\frac{\log\mathbf{p}^{(Y)}}{\mathbf{p}^{(Y)}}\|\nabla_\theta\log\mathbf{p}^{(Y)}\|_2^2\right]$ | 0.921 ($\pm$ 0.106) | Deep |
| $\mathbb{E}_{Y\sim\mathbf{p}}\left[\|\nabla_\theta\log\mathbf{p}^{(Y)}\|_1\right]$  (EXGRAD) | 0.925 ($\pm$ 0.047) | Shallow |
| $\mathbb{E}_{Y\sim\mathbf{p}}\left[\|\nabla_\theta\log\mathbf{p}^{(Y)}\|_1\right]$  (EXGRAD) | 0.926 ($\pm$ 0.053) | Deep |
| $\sum_{k=1}^C\mathbf{p}^{(k)}(1-\mathbf{p}^{(k)})$  (EXGRAD $V$ term) | 0.933 ($\pm$ 0.063) | *N/A* |
| $\sum_{k=1}^C\left\|\frac{1}{C}-\mathbf{p}^{(k)}\right\|$  (GRADNORM $V$ term) | 0.935 ($\pm$ 0.064) | *N/A* |

Table 1: **AUROC for Gradient-Based OOD Detectors on Small-Scale Experiments** The leftmost column shows the definition of each score function. The Gradient Depth column signifies whether gradients were taken with respect to all parameters ("Deep") or just the last layer's parameters ("Shallow"). For each row, the mean AUROC is reported with standard deviations in parentheses, both calculated across the 6 ID-OOD dataset combinations from {MNIST, CIFAR-10, SVHN}. Rows are sorted by mean AUROC. Note that the last two rows are the highest performing score functions and correspond to scores that do not involve any gradient calculations.

| Dataset | | $V$ Only | | | | $\|\mathbf{h}\|_1 V$ | | | Gradient-Based | |
|---|---|---|---|---|---|---|---|---|---|---|
| ID | OOD | MSP | Energy | VarSum | TV | MSP | Energy | VarSum | GradNorm | ExGrad |
| MNIST | CIFAR10 | 0.971 | 0.957 | 0.972 | **0.974** | 0.907 | 0.928 | 0.867 | 0.896 | 0.963 |
| | SVHN | 0.987 | 0.989 | 0.988 | **0.990** | 0.978 | 0.980 | 0.949 | 0.971 | 0.975 |
| CIFAR10 | MNIST | 0.924 | 0.831 | 0.925 | **0.928** | 0.913 | 0.860 | 0.895 | 0.913 | 0.921 |
| | SVHN | 0.943 | 0.926 | 0.944 | 0.948 | 0.994 | 0.957 | 0.995 | **0.995** | 0.919 |
| SVHN | MNIST | 0.813 | 0.807 | 0.813 | 0.811 | 0.748 | 0.740 | 0.637 | 0.735 | **0.840** |
| | CIFAR10 | 0.955 | 0.953 | **0.956** | 0.955 | 0.936 | 0.914 | 0.844 | 0.926 | 0.931 |
| Average | | 0.932 | 0.91 | 0.933 | **0.935** | 0.913 | 0.896 | 0.865 | 0.906 | 0.925 |
| ImageNet | iNaturalist | 0.876 | 0.885 | 0.884 | 0.885 | 0.892 | **0.917** | 0.748 | 0.904 | 0.769 |
| | SUN | 0.782 | 0.852 | 0.790 | 0.830 | 0.818 | **0.910** | 0.787 | 0.890 | 0.666 |
| | Places | 0.767 | 0.813 | 0.771 | 0.789 | 0.795 | **0.872** | 0.729 | 0.849 | 0.689 |
| | Textures | 0.744 | 0.758 | 0.750 | 0.761 | 0.778 | **0.817** | 0.729 | 0.811 | 0.651 |
| Average | | 0.792 | 0.827 | 0.799 | 0.816 | 0.821 | **0.879** | 0.748 | 0.864 | 0.694 |

Table 2: **AUROC scores** The table shows AUROC scores for several detectors grouped by category. The leftmost group uses only the output of the network ($V$ only), the second is the product of these measures with the 1-norm of the encoding passed to the last layer of the network $\|\mathbf{h}\|_1 V$, and the last grouping is detectors that leverage gradients of the last layer of the network. The highest AUROC found for each OOD task is bolded. We separately report the average of the ImageNet baselines and all of the other baselines. Note that we do show $\|\mathbf{h}\|_1 TV$ in this table since it is exactly GRADNORM.

10,000 images for each of the OOD datasets (except for Textures which uses 5,640). The pre-trained model is from Google BiT-S[2] (Kolesnikov et al., 2020) and uses the ResNetv2-101 architecture (He

---

[2] https://github.com/google-research/big_transfer

et al., 2016). Our code for these experiments was built on top of the code from Huang et al. (2021)[3], and more details about this baseline can be found in their paper.

**Experimenting with the Encoding-Output Composition** In what follows, we test a variety of detectors that adhere to the encoding-output composition described in Section 3.1. In particular, each of the methods we test here are a product of a $U$ term, which is a function of the encoding $\mathbf{h}$, and a $V$ term, which is a function of the outputted probability vector $\mathbf{p}$. We test all combinations of $U \in \{1, \|\mathbf{h}\|_1\}$ and $V \in \{\text{Energy, VarSum, MSP, TV}\}$.

Here, $U = 1$ denotes not using any encoding of the input and only using the output score (i.e. $V$ term). For the $V$ term, TV is the TV distance between $\mathbf{p}$ and a discrete uniform distribution (V term from Eq. 4). The scores for Energy (Liu et al., 2020) and MSP (Hendrycks & Gimpel, 2016) are as follows:

$$\mathcal{S}_{\text{Energy}} = T \log \sum_{k=1}^{C} e^{f^{(k)}(\mathbf{x})/T} \qquad \mathcal{S}_{\text{MSP}} = \max_{k \in [C]} \mathbf{p}^{(k)}$$

Note that the energy score we use here is the negative version of the one originally introduced in Liu et al. (2020). Like before, we assume that $T = 1$ for all experiments.

VarSum is a term inspired by the decomposition of EXGRAD (i.e the $V$ term from Eq. 5). Because $V = \sum_{k=1}^{C} \mathbf{p}^{(k)}(1 - \mathbf{p}^{(k)})$ is anti-correlated with every other score, we make VarSum a correlated version of this. In particular, VarSum $= 1 - \sum_{k=1}^{C} \mathbf{p}^{(k)}(1 - \mathbf{p}^{(k)})$. Each of the $V$ terms explored here are visualized in Figure 1, with the exception of Energy, since logits cannot be deduced from probabilities alone.

Table 2 displays the OOD detection performance for each of these methods in both the small-scale benchmark setting (with MNIST, CIFAR10, SVHN datasets) and the large-scale benchmark setting with the ImageNet dataset. The small-scale experiments assume the same setting as Section 4.1.

**Analysis of Results** It is immediately clear that there is a significant difference between the ImageNet baseline and the other baselines, and as such, we average the AUROCs separately. Starting with the small-scale setting, we find the best performers to be methods which look at the outputted predicted distribution only. Interestingly, the two best scores appear to be VarSum and the TV distance, which to the best of our knowledge, have not been recommended for OOD detection by previous works.

Literature generally warns against exclusively using the model's predicted distribution for OOD detection (Hein et al., 2019; Kirsch et al., 2021) since even OOD points can produce confident predictions with probabilities that are highly concentrated on a single class. These results indicate that this issue is milder in smaller scale models (MNIST, SVHN), and exacerbated for large models which are highly over-parameterized (CIFAR10, ImageNet). This result is generally in line with observations in calibration, which point out the over-confidence of neural network models as they become more over-parameterized (Guo et al., 2017; Wang et al., 2021).

Following this, it seems that information about the penultimate layer encoding is important for high performing encoders in the ImageNet baselines. Besides VarSum, every possible $V$ improves across all ImageNet baselines when multiplied by $\|\mathbf{h}\|_1$. It is unclear why VarSum and EXGRAD do not follow this trend and have subpar performance; however, we believe it may be related to how the sum of variance landscape changes with a dramatic increase in classes.

We believe that at the time of writing this paper GRADNORM is the current state-of-the-art post hoc method for all considered benchmarks. *However, importantly, we find that $\|\mathbf{h}\|_1 \times$ ENERGY is a strictly better detector than* GRADNORM *on the ImageNet benchmark.* These results further strengthen our claim that gradients do not necessarily provide unique benefits for OOD detection performance. It is perhaps only the encoding-output decomposition that results in strong performance, rather than some unique property of gradients.

### 4.3 EXPLORING ENCODING CHOICES

While the previous section limited the feature encoding to the $L_1$ norm, in this section, we study the effect of encoding design choice by varying the norm applied. Although Huang et al. (2021) do an

---

[3]https://github.com/deeplearning-wisc/gradnorm_ood

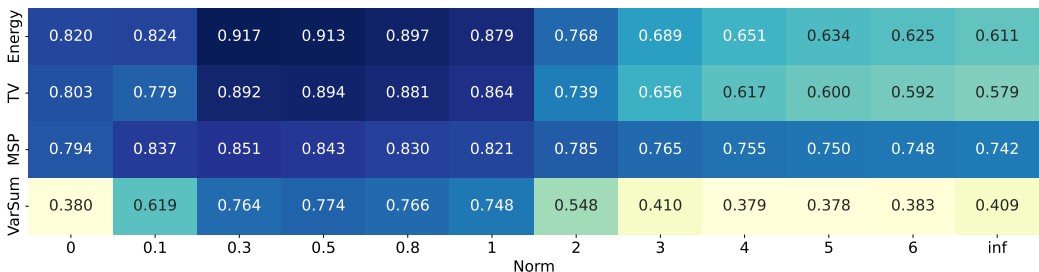

Figure 2: **Average AUROC over ImageNet Benchmark** Each cell of the heatmap shows the average AUROC for a different configuration of probability output score (y-axis) and order of the norm on the encoding fed to the last layer (x-axis).

ablation which changes the order of the norm, they do this ablation on the gradients themselves. By focusing on the norm of **h**, we are able to have more control over the score and the resulting detector.

Figure 2 displays the average AUROC performance in the ImageNet benchmark for 48 different pairs of $UV$ scores, where $U \in \{\|\mathbf{h}\|_p : p \in \{0, 0.1, 0.3, 0.5, 0.8, 1, 2, 3, 4, 5, 6, \infty\}\}$, and $V \in \{$Energy, TV, MSP, VarSum$\}$. Heatmaps for every experiment are shown in Appendix C. The results indicate that the choice of encoding does have a significant effect in a score's OOD detection performance. In particular, $\|\mathbf{h}\|_{0.3} \times$ Energy achieves an AUROC of 0.917, which is a drastic improvement over any score in Table 2. It is also better than some previously proposed non-post hoc methods such as MOS (Huang & Li, 2021), which achieves an average AUROC of 0.901.

We note that this result is not exactly fair since the parameter scan was done over the benchmark test set. Nevertheless, this is an encouraging result for what could possibly be achieved by a method that considers both image encoding and network output. In particular, we believe that devising more sophisticated $U$ terms than the naive ones tested here could be a promising direction for improving OOD detection performance within this framework.

### 4.4 CHALLENGING THE FEATURE-EXTRACTION HYPOTHESIS

As described above, one emerging theory that attempts to explain the role of gradients in OOD detection is the Feature-Extraction Hypothesis. Quoting Lee & AlRegib (2020),

> *Gradient-based optimization involves larger updates when there is a larger gap between predictions and correct labels for given inputs. It implies that the model requires more significant adjustments to its parameters, as it has not learned enough features to represent the inputs or relationships between learned features and classes for correct prediction.*

We challenge the plausibility of the Feature-Extraction Hypothesis in explaining the role of gradients with the following observation: the gradient calculated at a singleton test point $\tilde{\mathbf{x}}$ can be associated with a learning problem that requires *no* feature learning in order to minimise the loss. In particular, we note that previous gradient based scores involve calculating gradients for optimisation problems of the form $\min_\theta \mathbb{E}_Y[\ell(Y, \mathbf{p}(\tilde{\mathbf{x}}))]$. Notably, degenerate mappings[4] lie in the solution space for such optimisation problems, which by definition cannot extract features from input data. In order to investigate the plausibility of the Feature Extraction Hypothesis, we define a score function that by construction excludes degenerate mappings in its solution space. Specifically, we define the score

$$\mathcal{S}_{BG}(\mathbf{x}) = \mathbb{E}_{Y \sim \mathbf{p}} \left[ \left\| \nabla_\theta \left( \log \mathbf{p}^{(Y)}(\mathbf{x}) + \sum_{i=1}^{C} \log \mathbf{p}^{(Y_i^{\mathrm{ID}})}(\mathbf{x}_i^{\mathrm{ID}}) \right) \right\|_1 \right], \qquad (6)$$

where $(\mathbf{x}_i^{\mathrm{ID}}, Y_i^{\mathrm{ID}})$ is an in distribution training point belonging to class $i$. We refer to the OOD classifier based on this score function as BATCHGRAD. In particuar, we note that the loss function inside the parentheses of equation 6 defines an optimisation problem that, for sufficiently expressive networks, cannot have degenerate mappings as its solution. This is because the 2nd term inside

---

[4]Note that by "true mapping", we mean a mapping that is not degenerate, and by "degenerate mapping" we mean a mapping that maps to the same value for all inputs.

|  | Deep BATCHGRAD | Shallow BATCHGRAD |
|---|---|---|
| **AUROC** | 0.787 ($\pm$0.224) | **0.925** ($\pm$0.044) |

Table 3: **BATCHGRAD Small-Scale Experimental Results** This table shows AUROC results, calculated across the 6 ID-OOD dataset combinations from {MNIST, CIFAR-10, SVHN}.

the parentheses penalizes choices of $\theta$ that parameterise a degenerate mapping. Note that this is in contrast to other loss functions, such as those used in EXGRAD and GRADNORM, where degenerate mappings are in the solution spaces of their associated optimisation problems. The motivation for the BATCHGRAD loss function is to design an experiment where gradients are guaranteed to point towards a true (i.e. non-degenerate) mapping, giving a better chance of observing the Feature Extraction Hypothesis in action, should it be true. In particular, if the Feature Extraction Hypothesis were true, then we would be especially likely to see a reduction in OOD detection performance when restricting gradients to the final layer of BATCHGRAD. However, as shown in the results in Table 3 we find that the *opposite* holds: the variant of BATCHGRAD with gradients restricted to just the last layer is more informative for OOD detection than when using the gradients w.r.t. the whole network. This observation is consistent with previous experiments showing the sufficiency, and at times improvement, of restricting gradients to final layer parameters. This experimental result, in combination with analysis demonstrating the significance of the last layer gradients, leads us to conclude that the Feature-Extraction Hypothesis is not an appropriate explanation for the high performance of gradient-based OOD detection methods.

## 5 DISCUSSION

In this work, we experimentally investigated gradient-based OOD detection methods. While using gradient-based approaches can result in strong performance, we find previous explanations attributing performance to gradients unsatisfactory. Although prior works have focused on the interpretation of taking the gradient with respect to the KL divergence between the outputted distribution and a discrete uniform distribution (Lee & AlRegib, 2020; Huang et al., 2021), we find that other gradient-based scores that do not have this interpretation *also perform well*, especially on smaller scale problems.

We also question the idea that the gradient space of the neural network holds key information used for OOD detection. Our experiments provide evidence against the hypothesis that gradient-based methods are informed by large changes needed in the network to capture unseen, OOD images. Moreover, we show that we can derive better performing detectors that are agnostic to gradients and only use the encoding-output decomposition discussed in Huang et al. (2021). As such, we believe the strength of GRADNORM comes not from its leverage of gradients, but solely from the fact that it fuses information about network encodings and outputted distributions. *Hence, while it is possible that gradients contain useful information for OOD detection, we do not believe that previous methods leverage information from gradients that cannot be derived more easily through other means.*

**Future Work** Our hope is that the insights provided in this work can be used to further improve OOD detection. In particular, we believe that more advanced methods can be developed that fuse together information from the network encoding and scores derived from the model's predicted distribution. For example, we evaluated the choice of input encodings by varying the order of the norm applied on the last hidden layer outputs. Investigating the utility of other hidden layer features or auxiliary encoders and studying what properties of an encoding are helpful in detecting OOD data could provide further insights to devising stronger OOD detectors.

Another interesting direction for investigation is what role task difficulty and model capacity play in how these detectors perform. We found that when the in distribution task was easier, top performing detectors only depended on network outputs; however, it was essential to use both outputs and input encodings for the more complex ImageNet tasks. As mentioned in Section 4.2, existing works in uncertainty quantification and calibration have noted the unreliability of a deep model's predicted class distribution. While calibration is orthogonal to the scope of this work, we believe there could be fundamental ties in determining the boundary between solely relying on the output predicted distribution for OOD detection (i.e. utilizing the $V$ term only), and additionally requiring extracted information from the input space via input encodings (utilizing the $U$ term). We leave investigating this direction for future work.

**Reproduciblity Statement** Our experimental setup follows closely that of our main baseline Huang et al. (2021). We have followed their code as provided in the public github repo `https://github.com/deeplearning-wisc/gradnorm_ood`, which includes the pre-trained models and data processing. For the "small scale" experiments (any experiments involving MNIST, CIFAR10, SVHN datasets), we use the pre-trained models as they are provided in the public github repo `https://github.com/aaron-xichen/pytorch-playground`, and follow the rest of the protocol as provided in the earlier repo.

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

## A  ADDITIONAL EXPERIMENTAL DETAILS

The tables below shows detailed breakdowns of AUROC results for the small-scale experiments. Note that the top row describes the ID dataset, and the two entries beneath describe the OOD datasets.

| Score Expression | Gradient | SVHN | | MNIST | | CIFAR | |
|---|---|---|---|---|---|---|---|
| | | MNIST | CIFAR | SVHN | CIFAR | SVHN | MNIST |
| $\left\|\mathbb{E}_{Y\sim\mathbf{p}}\left[\nabla_\theta\log\mathbf{p}^{(Y)}\right]\right\|_2^2$ | Deep | 0.595 | 0.730 | 0.839 | 0.786 | 0.669 | 0.733 |
| $\left\|\mathbb{E}_{Y\sim\mathbf{p}}\left[\nabla_\theta\log\mathbf{p}^{(Y)}\right]\right\|_2^2$ | Shallow | 0.651 | 0.796 | 0.838 | 0.775 | 0.635 | 0.749 |
| $\mathbb{E}_{Y\sim\text{Uniform}}\left[\left\|\nabla_\theta\log\mathbf{p}^{(Y)}\right\|_1\right]$ | Deep | 0.692 | 0.816 | 0.790 | 0.717 | 0.993 | 0.940 |
| $\mathbb{E}_{Y\sim\text{Uniform}}\left[\left\|\nabla_\theta\log\mathbf{p}^{(Y)}\right\|_1\right]$ | Shallow | 0.606 | 0.817 | 0.939 | 0.857 | 0.995 | 0.886 |
| $\mathbb{E}_{Y\sim\text{Uniform}}\left[\left\|\nabla_\theta\log\mathbf{p}^{(Y)}\right\|_2^2\right]$ | Deep | 0.591 | 0.895 | 0.986 | 0.957 | 0.954 | 0.819 |
| $\mathbb{E}_{Y\sim\text{Uniform}}\left[\left\|\nabla_\theta\log\mathbf{p}^{(Y)}\right\|_2^2\right]$ | Shallow | 0.709 | 0.919 | 0.976 | 0.903 | 0.994 | 0.819 |
| $\left\|\nabla_\theta\mathbb{E}_{Y\sim\text{Uniform}}\left[\log\mathbf{p}^{(Y)}\right]\right\|_1$ | Deep | 0.759 | 0.937 | 0.911 | 0.814 | 0.989 | 0.943 |
| $\left\|\nabla_\theta\mathbb{E}_{Y\sim\text{Uniform}}\left[\log\mathbf{p}^{(Y)}\right]\right\|_1$ | Shallow | 0.735 | 0.926 | 0.971 | 0.896 | 0.995 | 0.913 |
| $\mathbb{E}_{Y\sim\mathbf{p}}\left[\frac{\log\mathbf{p}^{(Y)}}{\mathbf{p}^{(Y)}}\left\|\nabla_\theta\log\mathbf{p}^{(Y)}\right\|_2^2\right]$ | Shallow | 0.749 | 0.933 | 0.983 | 0.928 | 0.994 | 0.870 |
| $\mathbb{E}_{Y\sim\mathbf{p}}\left[\left\|\nabla_\theta\log\mathbf{p}^{(Y)}\right\|_2^2\right]$ | Shallow | 0.848 | 0.908 | 0.964 | 0.955 | 0.915 | 0.925 |
| $\mathbb{E}_{Y\sim\mathbf{p}}\left[\left\|\nabla_\theta\log\mathbf{p}^{(Y)}\right\|_2^2\right]$ | Deep | 0.862 | 0.936 | 0.956 | 0.947 | 0.916 | 0.910 |
| $\mathbb{E}_{Y\sim\mathbf{p}}\left[\frac{\log\mathbf{p}^{(Y)}}{\mathbf{p}^{(Y)}}\left\|\nabla_\theta\log\mathbf{p}^{(Y)}\right\|_2^2\right]$ | Deep | 0.710 | 0.926 | 0.993 | 0.968 | 0.984 | 0.947 |
| $\mathbb{E}_{Y\sim\mathbf{p}}\left[\left\|\nabla_\theta\log\mathbf{p}^{(Y)}\right\|_1\right]$ | Shallow | 0.840 | 0.931 | 0.975 | 0.963 | 0.919 | 0.921 |
| $\mathbb{E}_{Y\sim\mathbf{p}}\left[\left\|\nabla_\theta\log\mathbf{p}^{(Y)}\right\|_1\right]$ | Deep | 0.830 | 0.947 | 0.977 | 0.965 | 0.921 | 0.918 |
| $\sum_{k=1}^C\mathbf{p}^{(k)}(1-\mathbf{p}^{(k)})$ | *N/A* | 0.813 | 0.956 | 0.988 | 0.972 | 0.944 | 0.926 |
| $\sum_{k=1}^C\left\|\frac{1}{C}-\mathbf{p}^{(k)}\right\|$ | *N/A* | 0.811 | 0.955 | 0.990 | 0.974 | 0.948 | 0.928 |

| ID Dataset | SVHN | | MNIST | | CIFAR | |
|---|---|---|---|---|---|---|
| OOD Dataset | MNIST | CIFAR | SVHN | CIFAR | SVHN | MNIST |
| Deep BATCHGRAD | 0.843 | 0.947 | 0.974 | 0.950 | 0.505 | 0.502 |
| Shallow BATCHGRAD | 0.851 | 0.922 | 0.975 | 0.963 | 0.920 | 0.921 |

## B  DERIVATION OF EXGRAD DECOMPOSITION

We will now rewrite Eq. 3 into a more digestible form. Assume that we only take the derivative with respect to the last layer of the network, assume that $W\in\mathbb{R}^{C\times D}$ are the parameters for the last layer, and let the output of the last layer be $f_\theta(\mathbf{h})=W\mathbf{h}$, where $\mathbf{h}$ is the encoding taken in by the last layer. Although we have not considered bias terms in this formulation, note that they can easily be included by adding another dimension to $\mathbf{h}$ with the value 1. For readability, we write $\mathbf{p}$ instead of $\mathbf{p}(\mathbf{h})$ and $f^{(k)}$ instead of $f^{(k)}(\mathbf{h})$.

First, fix a class $k$, and note that

$$\frac{\partial}{\partial f_\theta^{(k)}}\log\left(\frac{e^{f_\theta^{(k)}/T}}{\sum_{k'=1}^C e^{f_\theta^{(k')/T}}}\right)=\frac{\partial}{\partial f_\theta^{(k)}}\left(\frac{f_\theta^{(k)}}{T}-\log\sum_k e^{f_\theta^{(k')}/T}\right)$$

$$=\frac{1}{T}\left(1-\frac{e^{f_\theta^{(k)}}}{\sum_k e^{f_\theta^{(k')}}}\right)$$

$$=\frac{1}{T}\left(1-\mathbf{p}^{(k)}\right)$$

For a class $k' \neq k$,

$$\frac{\partial}{\partial f_\theta^{(k')}} \log \left( \frac{e^{f_\theta^{(k)}}}{\sum_{k'=1}^C e^{f_\theta^{(k')}}} \right) = -\frac{e^{f_\theta^{(k')}}/T}{\sum_k e^{f_\theta^{(k')}}/T}$$
$$= \frac{-\mathbf{p}^{(k')}}{T}$$

Building on this,

$$\frac{\partial}{\partial W} \log \left( \frac{e^{f_\theta^{(k)}/T}}{\sum_{k'=1}^C e^{f_\theta^{(k')}/T}} \right) = \frac{1}{T}\mathbf{h} \left[ -\mathbf{p}^{(1)}, \quad -\mathbf{p}^{(2)}, \quad \ldots, \quad 1 - \mathbf{p}^{(k)}, \quad \ldots \right]$$
$$= \frac{1}{T} \begin{bmatrix} -\mathbf{p}^{(1)}\mathbf{h}^{(1)} & -\mathbf{p}^{(2)}\mathbf{h}^{(1)} & \ldots & (1-\mathbf{p}^{(k)})\mathbf{h}^{(1)} & \ldots \\ -\mathbf{p}^{(1)}\mathbf{h}^{(2)} & -\mathbf{p}^{(2)}\mathbf{h}^{(2)} & \ldots & (1-\mathbf{p}^{(k)})\mathbf{h}^{(2)} & \ldots \\ \vdots & \vdots & \vdots & \vdots & \vdots \end{bmatrix}$$

The $L1$ norm of the gradient with respect to $W$ is just the sum of the absolute values of each entry.

$$\sum_{i=1}^D \left( (1-\mathbf{p}^{(k)})|\mathbf{h}_i| + \sum_{k'\neq k} \mathbf{p}^{(k')}|\mathbf{h}_i| \right) = \sum_{i=1}^D \left( (1-\mathbf{p}^{(k)})|\mathbf{h}_i| + (1-\mathbf{p}^{(k)})|\mathbf{h}_i| \right)$$
$$= 2(1-\mathbf{p}^{(k)}) \|\mathbf{h}\|_1$$

Putting it all together,

$$S(x) = \mathbb{E}_{k\sim\mathbf{p}} \left[ \left\| \nabla_W \log \left( \frac{e^{f_\theta^{(k)}/T}}{\sum_{k'=1}^C e^{f_\theta^{(k')}/T}} \right) \right\|_1 \right]$$
$$= \frac{2}{T}\mathbb{E}_{k\sim\mathbf{p}} \left[ (1-\mathbf{p}^{(k)}) \|\mathbf{h}\|_1 \right]$$
$$= \frac{2}{T} \|\mathbf{h}\|_1 \sum_{c=1}^C \mathbf{p}^{(k)}(1-\mathbf{p}^{(k)})$$

## C  ADDITIONAL HEATMAPS

In this section, we present additional heatmaps (similar to Figure 2) which indicate the AUROC for each of the methods evaluated in Section 4.3, separately for each of the ID-OOD dataset settings tested in the experiments section. The title of each plot indicates the ID and OOD dataset in order, i.e. "ID x OOD". Each cell of the heatmap shows the AUROC for a different configuration of probability output score (y-axis) and order of the norm on the encoding fed to the last layer (x-axis).

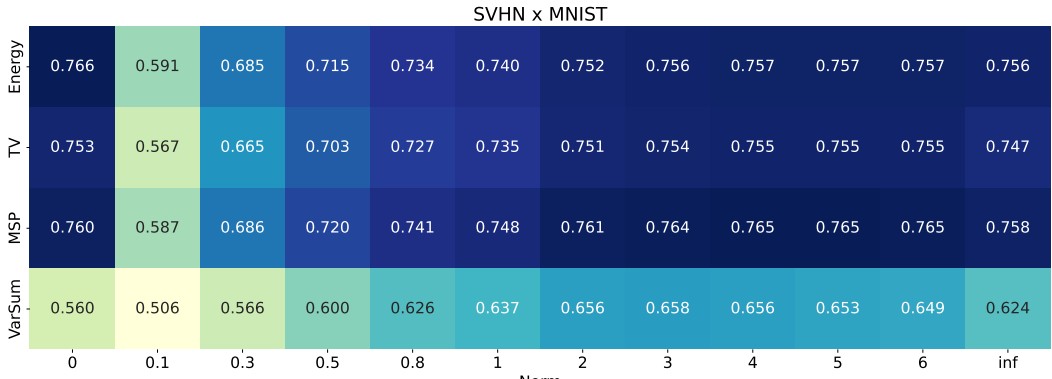

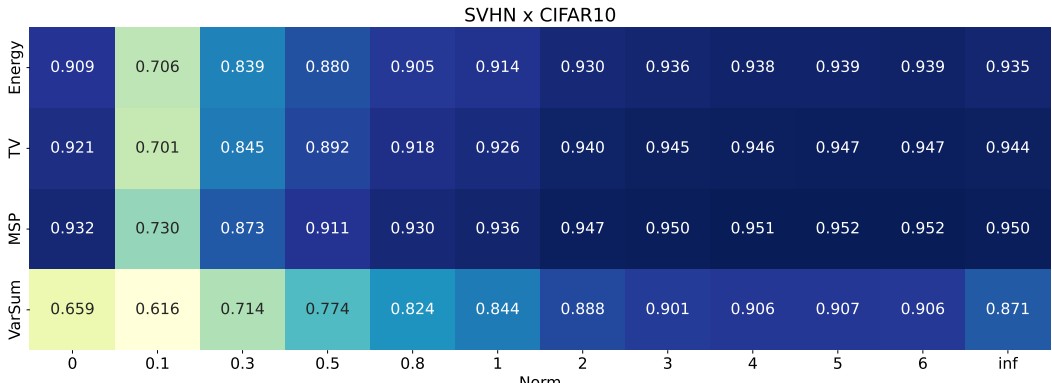

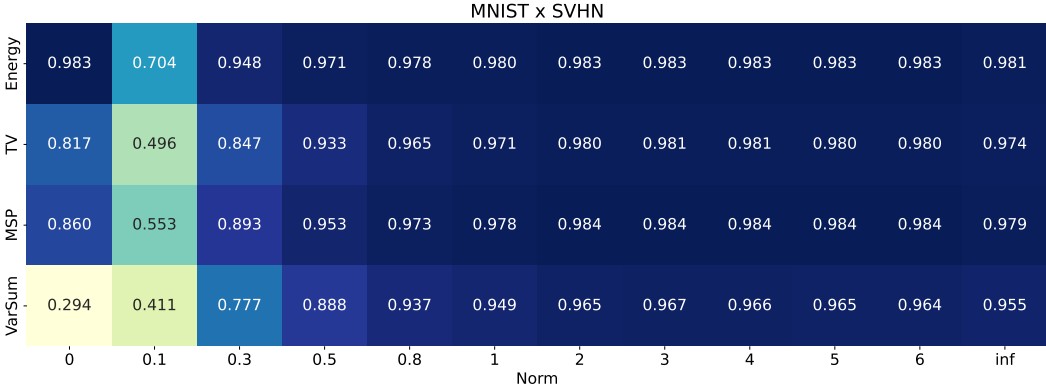

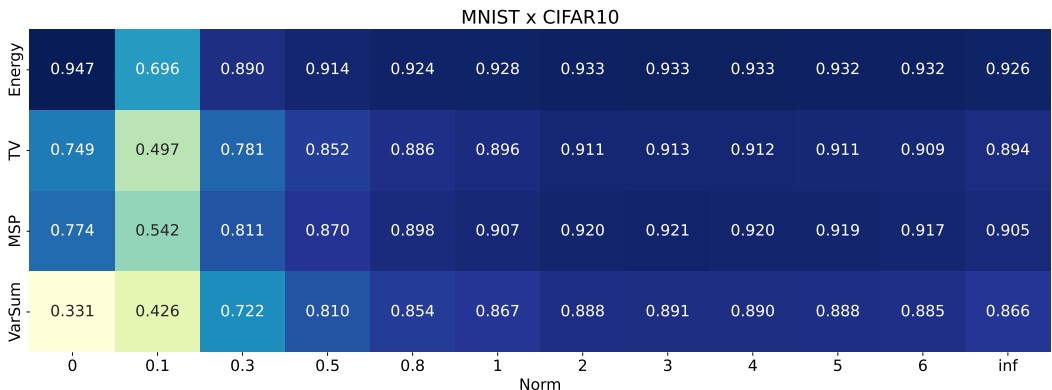

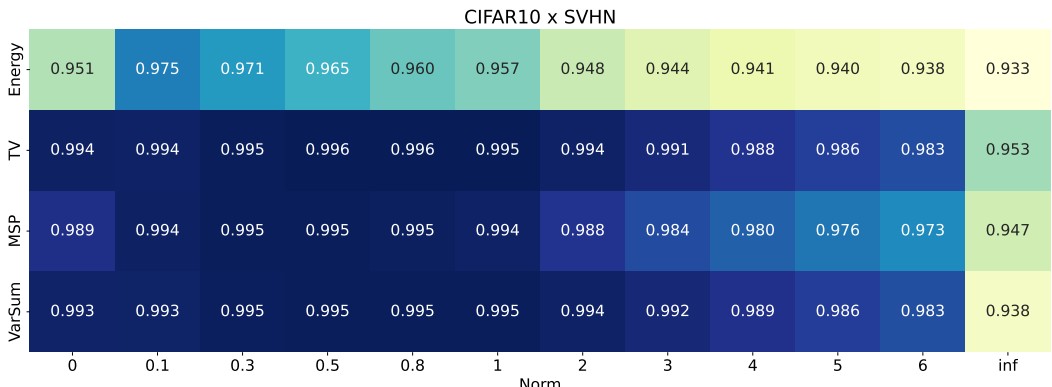

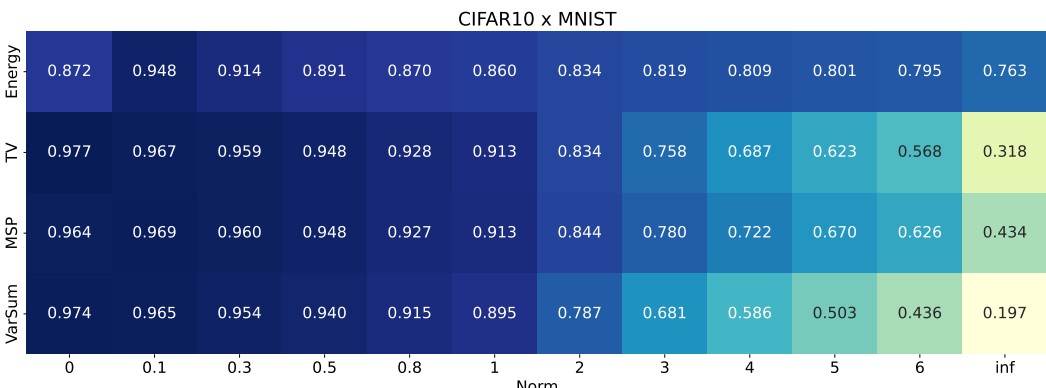

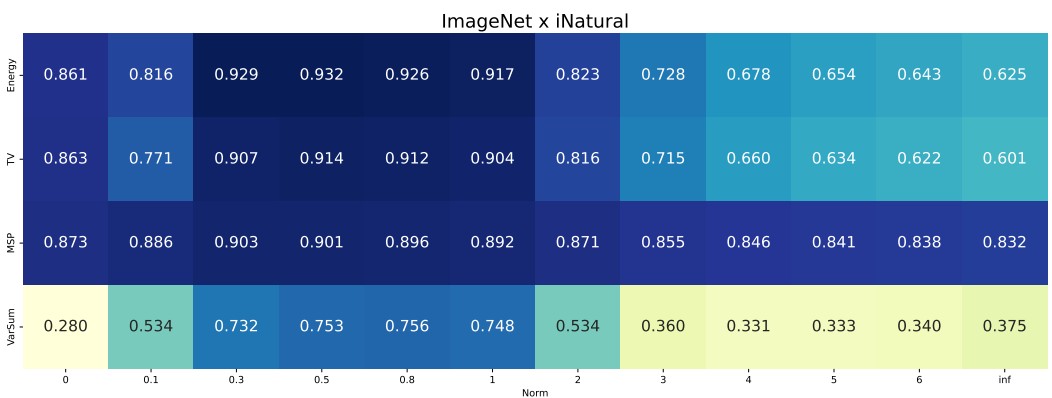

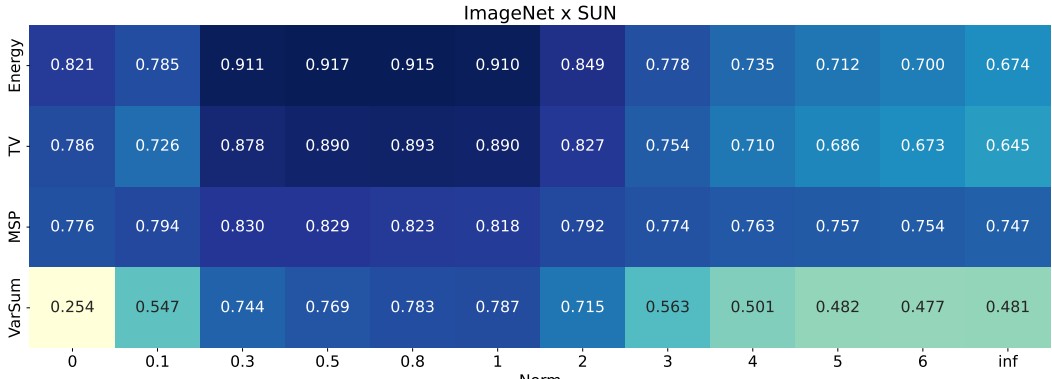

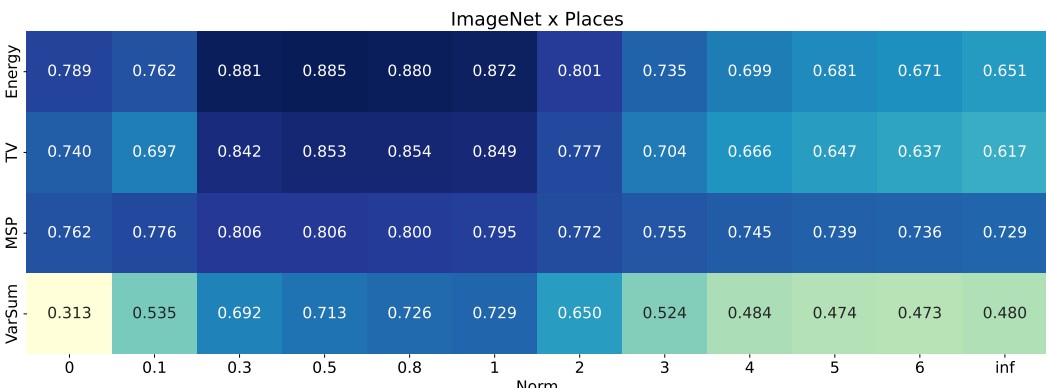

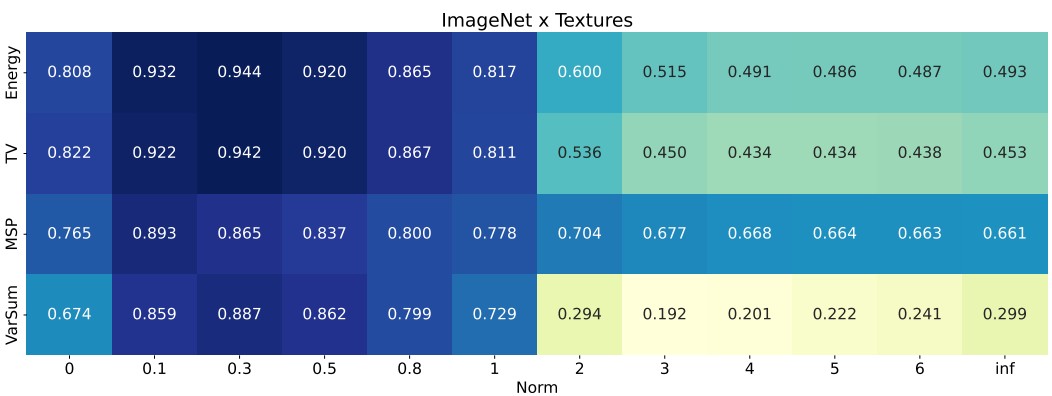

