# OpenReview forum: "How Useful are Gradients for OOD Detection Really?"
_ICLR.cc/2023/Conference — Submitted to ICLR 2023_

### Official Review · Reviewer_RRxz · 2022-10-21

**Confidence:** 4
**Clarity, Quality, Novelty And Reproducibility:** See Strength And Weaknesses.
**Correctness:** 3
**Technical Novelty And Significance:** 2
**Empirical Novelty And Significance:** 2
**Recommendation:** 5

**Strength And Weaknesses:**


Strength:

1. This is the first paper systematically explore the role of different components for gradient-based methods in OOD detection.
2. Some new insights are provided like the V component in GradNorm can be regarded as total variation and the V component in ExGrad can be regarded as the variance of a Bernoulli variable.
3. The related work is extensive.

Weakness:

1. The paper does not offer a finding that is interesting and surprising enough to readers. Several observations are listed in a scattered way, which also makes the paper distractive. For example:

	a) The author finds that |h|_1 × ENERGY is a strictly better detector than GRADNORM. My feeling is that you can do any recombinations of different OOD scoring functions and find a specific combination (B + D) that performs specifically well. However, such a finding is very likely due to an "overfitting" of the test setup. As evidence, |h|_1 × ENERGY does not perform well on small-scale settings as the author showed.

	b) The author claims that a strict derivation from the gradient form is not needed. That is true, but not surprising. The GradNorm is simply a combination of U and V as pointed out in the GradNorm paper. Both U and V have the discriminative power for ID vs OOD. You can change U or V to other discriminative scoring functions and can still work. A more interesting question is that why U and V synthetically work better than U and V alone.

	c) The author claims that the Feature-Extraction Hypothesis is not applicable to the explanation of GradNorm. That is also understandable because they are in nature, not conflict. GradNorm is working based on the assumption for input sample x alone: OOD data are closer to a more flat probability space. Feature-Extraction Hypothesis is applied on both input x and label y: if you offer the input sample with a wrong label, the gradient will be large. I don't see why it is essentially counterintuitive.

2. The writing needs to be improved especially in Section 3 and Section 4.


**Summary Of The Paper:**


This paper investigates the role of gradient-based methods in OOD detection. Specifically, the author tries to answer the following questions and provide answers correspondingly:

1. Is it essential to use the scoring function strictly derived from the gradient?
No. The recombination of different components also works fine.

2. How do we align the feature-extraction hypothesis with the strong performance of gradient-based OOD detection methods?
Feature-Extraction Hypothesis is not an appropriate explanation.

3. Does gradient-based approaches offer unique performance advantages for post hoc OOD detection?
No. Essentially the same reason as question 1.

**Summary Of The Review:**

The paper is performing an extensive and in-depth study of the gradient-based method. However, I don't share the excitement upon reading the finding provided in the paper, since I find the observation listed is common and not surprising. Therefore I lean toward rejecting the paper at this point.

---

> ### Author Response · Authors · 2022-11-16
> **Response to Reviewer RRxz**
>
> Thank you for your review. We appreciate your feedback.
>
> Addressing weakness 1. a, b) Our intention was never to champion a single method for OOD detection; rather, our goal was to explore the various scores that are possible with a $UV$ combination and show that they can be performant without requiring expensive gradient computation. It is true that a subset of our results may be overfitted (which we were careful to highlight in the paper). However, we believe that overall our results are nonetheless informative for guiding future OOD detection research. We also respectfully disagree with the claim that "[our] paper does not offer a finding that is interesting and surprising enough to readers". Since the publication of the original two gradient-based OOD papers from Huang et al (2021) and Lee et al (2020), there has been a small flurry of interest within the OOD detection community, with several recent works proposing gradient-based OOD methods building on the original two works [1-9].
>
> Addressing weakness c) We disagree that GradNorm does not make use of label information. In fact, GradNorm does indeed require a label: the uniform label. The motivation behind their method (which is re-stated in Section 3, in the paragraph immediately below Hypothesis 3.1) is that taking a KL of the output distribution w.r.t. a uniform label will necessitate higher gradients to the model parameters for ID data. Hence, the feature-extraction hypothesis bears relevance to GradNorm.
>
> Addressing weakness 2. We have substantially updated the writing in section 4 to improve clarity (changes shown in teal). Would you be willing to provide further comments on exactly how the writing can be further improved? All other reviewers have noted the clarity and high quality of the writing, so it would be helpful if you could specify your concerns here.
>
> [1] Gradient-Based Novelty Detection Boosted by Self-supervised Binary Classification, J. Sun et al., 2022
>
> [2] R2-AD2: Detecting Anomalies by Analysing the Raw Gradient, JP Schulze et al., 2022
>
> [3] Gradient-Based Severity Labeling for Biomarker Classification in OCT, K. Kokilepersaud et al., 2022
>
> [4] Self-supervised Novelty Detection for Continual Learning: A Gradient-Based Approach Boosted by Binary Classification, J. Sun et al., 2021
>
> [5] Gradient-Based Uncertainty for Monocular Depth Estimation, J Hornauer et al., 2022
>
> [6] Open-Set Recognition With Gradient-Based Representations, J. Lee et al., 2022
>
> [7] Outlier-Robust Group Inference via Gradient Space Clustering, Y. Zeng, 2022
>
> [8] Gradient-Based Adversarial and Out-of-Distribution Detection, J. Lee et al., 2022
>
> [9] Explanatory Paradigms in Neural Networks: Towards Relevant and Contextual Explanations, G. AlRegib et al., 2022

---

### Official Review · Reviewer_ipBL · 2022-10-23

**Confidence:** 4
**Correctness:** 3
**Technical Novelty And Significance:** 2
**Empirical Novelty And Significance:** Not applicable
**Recommendation:** 5

**Clarity, Quality, Novelty And Reproducibility:**

Clarity: The writing and exposition is very clear.

Quality: In my view, the significance of this work does not meet the bar for ICLR.

Novelty: There is no significant novelty in the paper; the choices for U and V, and norm-orderings are largely heuristic, and derived from existing intuitions.

Reproducibility: There are adequate details in the submission for meaningful reproducibility.

**Strength And Weaknesses:**

Strengths:

Huang et al. had discussed how the gradient-based method might work, through a product of two terms that are to do with the predictive distribution and the latent encoding separately. The submission performs a fairly detailed analysis of possible alternatives to these terms, not necessarily decomposed from a gradient-based score. The conclusions are sensible — gradient-based methods don’t seem essential to high performance, and alternatives can be formed with different heuristic choices for an OOD score. The paper is very well-written, which made it a pleasure to read.

Weaknesses:

My main comment is about relevance and significance to ICLR. The submission claims that GradNorm is SOTA at the time of writing, which would justify investing effort in rigorously analyzing the method, exposing limitations, and suggesting alternatives. However, there seem to be at least a couple other papers with equally good and better numbers around, for example, [1, 2, 3], all published reasonably earlier than the ICLR deadline (not all of them report all datasets in the submission, but without an evaluation, one cannot disregard them). These methods are not gradient-based, and seem to be well-aware that gradient-based methods aren’t the best way to go. In general, the narrative in the submission makes it appear as if the community is currently of the widespread opinion that gradient-based methods are top-choice, which needs to be corrected. In my view, past and contemporary literature does not indicate this, thus making the contributions of this submission less significant. There are, to my knowledge, only the two works of note (in the ocean of papers on OOD detection) that use gradients — Huang et al. (2021), and Lee and alRegib (2020). Lee and alRegib has been discussed at length in Huang et al., in particular the point that training a binary classifier on the test-distributions is poor methodology, and this paper is generally not well-cited — I only mention this to counter the narrative in the submission that there is an “emerging theory” in the literature that needs challenging (at a top-tier venue), with substantial portions dedicated to debunking the “feature-extraction hypothesis”. In my opinion, this sort of challenge is best-suited for a “Letter to the Editor” section, as in other scientific journals; unfortunately, our field does not seem to have allocated space for similar back-and-forth between author and challenger.

Given that no method is universally better, one cannot really draw any strong conclusions. Using a network’s “properties” (activation strengths and predictive distributions) implies that results are strongly dependent on the network and training choices. Using different underlying choices might change trends significantly. This is not a problem unique to this submission, but it somewhat comes in the way of the sort of analysis it seeks to provide: reporting numbers for several different modifications of the score might be reporting results that are “overfitted” on the specific underlying frameworks. The fact that the new alternative, ExGrad (and VarSum variants) underperform on the ImageNet benchmarks also hurts what might have been a positive point for technical novelty.


[1] ReAct: OOD Detection with Rectified Activations, NeurIPS 2021

[2] ViM: Out-Of-Distribution with Virtual-logit Matching, CVPR 2022

[3] Out-of-Distribution Detection with Deep Nearest Neighbors, ICML 2022

**Summary Of The Paper:**

The submission primarily aims at refuting ideas expressed in the NeurIPS21 paper “On the importance of gradients for detecting distributional shifts in the wild” (Huang et al.), by suggesting that gradients are actually not intrinsically required for the high performance in that paper; it’s more to do with the nature of the contributing terms once the GradNorm score is viewed with the decomposition in Huang et al. While this analysis was already made in Huang et al., the submission proposes variants of the individual terms, showcasing improvements over GradNorm with alternative choices.

**Summary Of The Review:**

The submissions takes a thorough look at GradNorm, following up on the initial analysis provided in Huang et al. about the benefits lying in the separated components of the score. However, in my view, the analysis and main takeaway (that gradient-based scores are not a requisite for high performance; and one can look for other ways to develop OOD-scores using activations and predictive distributions) are less relevant given the plethora of high-performing non-gradient based methods that already take such messages to heart. Therefore, my recommendation at this time is rejection.

---

> ### Author Response · Authors · 2022-11-16
> **Response to Reviewer ipBL**
>
> Thank you for your review. We appreciate your feedback.
>
> While works [1, 2, 3] that you reference above indeed mention gradient-based methods, we disagree that these works are "well-aware that gradient-based methods aren’t the best way to go". None of these papers makes such an argument; their discussion simply notes them as a possible alternative. This leaves room for many in the community to believe that a more refined version of Huang et al. (2021) would lead to a new SOTA method. Indeed we found at least 8 papers published in the last 12 months either citing Huang et al (2021) or Lee et al (2020) that directly build off of the idea of using gradients for OOD detection [4-12]. In addition, 3 survey papers make reference to Gradient-Based OOD detection methods [13-15] (2 of which are already highly cited), and each survey paper cites both Huang et al. (2021) & Lee et al (2020) as examples of recent work. With so many new methods of OOD detection being published every year, investigative works such as ours are important for focusing the community’s attention on methods with the most potential. As such, we assert that the community has much to benefit from investigative works appearing at top conferences.
>
> Lastly, while no method is universally better than any other, we still believe our experiments give insight that will aid in the development of new methods in the future.
>
> [4] Gradient-Based Novelty Detection Boosted by Self-supervised Binary Classification, J. Sun et al., 2022
>
> [5] R2-AD2: Detecting Anomalies by Analysing the Raw Gradient, JP Schulze et al., 2022
>
> [6] Gradient-Based Severity Labeling for Biomarker Classification in OCT, K. Kokilepersaud et al., 2022
>
> [7] Self-supervised Novelty Detection for Continual Learning: A Gradient-Based Approach Boosted by Binary Classification, J. Sun et al., 2021
>
> [8] Gradient-Based Uncertainty for Monocular Depth Estimation, J Hornauer et al., 2022
>
> [9] Open-Set Recognition With Gradient-Based Representations, J. Lee et al., 2022
>
> [10] Outlier-Robust Group Inference via Gradient Space Clustering, Y. Zeng, 2022
>
> [11] Gradient-Based Adversarial and Out-of-Distribution Detection, J. Lee et al., 2022
>
> [12] Explanatory Paradigms in Neural Networks: Towards Relevant and Contextual Explanations,  G. AlRegib et al., 2022
>
>
> [13] A Survey of Uncertainty in Deep Neural Networks, J. Gawlikowski et al., 2021
>
> [14] Generalized Out-of-Distribution Detection: A Survey, J. Yang et al., 2021
>
> [15] OpenOOD: Benchmarking Generalized Out-of-Distribution Detection, J. Yang et al., 2022

---

> > ### Comment · Reviewer_ipBL · 2022-11-22
> > **Follow-up**
> >
> > Thanks for the response!
> >
> > When I said that the existing SOTA papers are “well-aware that gradient-based methods aren’t the best way to go”, I did not mean that they must have provided explicit arguments for why this is not the case. It is rather an implicit inference, given that these methods, coming after Huang et al., and improving numbers on it, do not choose to develop a gradient-based method. Surely, if it were a common belief that gradients are best for OOD detection, and if it worked best in practice, all of these papers would be making use of such information? As to the room left for people to provide improvements by refining Huang et al.’s method, it seems possible that such a refinement might involve improvements which the heuristics proposed in the submission would not achieve (one cannot really tell, based on the current level of evidence). Making claims about the futility of pursuing refinements to a line of thinking is tenuous, without something like an impossibility theorem. In my view, people are welcome to attempt beating SOTA numbers (which currently do not involve using gradients) using gradient information, and if they are successful, that would be positively reported. If they fail, gradient-based methods will fall out of favour.
> >
> > Thanks for providing the list of works referencing gradient-based OOD-detection papers! I took a quick look at them: [5] uses a temporal analysis of gradient evolution -- perhaps a temporal version can achieve things the heuristic replacements of U and V cannot? (One cannot really tell.) [7] is the same as [4]. [6,9,11,12] seem to be incremental follow-ups upon the original paper by Lee and alRegib, and do not seem like significant/influential papers (I’m only saying this because ICLR is considered a top conference in AI/ML/DL, handling matters of highest relevance, and a subjective assessment of community-relevance does play a role). [13,14,15] are review papers, and review papers need to be inclusive. In fact, [15] evaluates several methods under several settings, and the numbers appear to confirm that GradNorm isn’t the best method out there.
> >
> > Overall, I am still quite unconvinced that this question has the sort of relevance in the OOD detection community as suggested by the submission, although I am improving my rating. As I suggested initially, and as Reviewer RRxz also points out, multiple heuristic combinations of heuristic scores are likely to find some random overfitted number on some test sets (when using a unique combination of base architectures and training choices). Such findings are not unexpected, and does not really inform us of general and meaningful. This submission might be improved with more thorough empiricism, showing how all these heuristic choices perform over a wider range of base models and training choices than explored in the current draft.

---

### Official Review · Reviewer_YM1z · 2022-10-24

**Confidence:** 3
**Correctness:** 3
**Technical Novelty And Significance:** 3
**Empirical Novelty And Significance:** 4
**Recommendation:** 8

**Clarity, Quality, Novelty And Reproducibility:**

Overall, I found the paper enjoyable to read and found the argument easy to follow, with the paper well-structured. I have highlighted a few aspects above which I believe could do with some clarification.

**Strength And Weaknesses:**

Overall, I enjoyed reading this paper (though not all parts were entirely clear to me). This paper provides a clear and focussed investigation into the question posed in the title, which I believe is useful for researchers and practitioners in the OoD detection community.

Strengths:
* This paper provides an in-depth analysis of GradNorm and related gradient-based OoD detection methods, which have recently gained traction in the OoD community. The paper analyses GradNorm (a SoTA OoD detection method) and its variants through both an empirical and theoretical analysis to draw conclusions. They find that many choices of the 'V' term could be equally useful for identifying OoD samples, even ones that are not explicitly related to a gradient computation.
* The authors investigate a 'Feature Extraction Hypothesis', an intuitive assumption that the gradient of the cross-entropy loss (with a uniform label) for an OoD sample would high, as a large change in the features is required to model the OoD sample. They authors conduct empirical and theoretical analysis and find evidence *against* this claim.
* The authors show a promising direction forward for OoD detection, which focusses on the norm of the extracted features. Though tuning on the test set, the authors show that SoTA OoD detection on the ImageNet-scale evaluation can be achieved simply by tuning the P-norm of the extracted features, and combining with the Energy scoring rule.

Weaknesses:
* The authors heavily use Energy as scoring rule in the 'V' term, implicitly treating it as independent of the feature norm in U. However, the energy function is itself impacted by the feature norm. This should be discussed.
* Perhaps I have misunderstood, but the authors highlight the opposite intuitions behind GradNorm and the Feature Extraction Hypothesis. Specifically, one suggests that the gradient norm should be high for ID examples, while the other says it should be high for OoD samples. If this is the case, should one of them not perform very poorly for OoD detection (AUROC <<50, given that they are anti-correlated)? If this is the case, is the Feature Extraction hypothesis not trivially verifiable or falsifiable? Perhaps the authors could clarify.

Misc:
* [1] discusses the role of the feature norm in open-set recognition and OoD detection. They provide intuitions as to why low-feature norms may occur for OoD samples, and further suggest the maximum logit score (MLS) as an alternative to MSP for open-set scoring (as the former is sensitive to feature norm). The authors should discuss this.
* The authors have used different parameters with respect to which to take the gradient in Eqs 2 and 3. This makes things less clear, so authors should either change this or discuss the difference.

[1] Open-set Recognition: a Good Closed-Set Classifier is All You Need?, Vaze et al., ICLR 22

**Summary Of The Paper:**

This paper questions a recent assumption in the OoD detection literature: that gradient information is useful for OoD detection. Specifically, the authors break down the norm of the gradient as 'U * V' (following the GradNorm paper, a SoTA OoD detection method), where U represents the feature norm and V is some function of the output layer.

The authors conduct both theoretical and empirical analysis under this framework to conclude that the gradient information provides little extra information that could not be extracted in another way. In particular, they find that the feature norm plays an important role in the high performance of the GradNorm method. They also develop a new scoring rule, ExGrad, which empirically outperforms GradNorm on small-scale OoD detection benchmarks, and provide analysis on both large and small-scale OoD detection benchmarks.

**Summary Of The Review:**

This is a good paper which provides clear and focussed analysis into a narrow research question. I think it will be useful to the OoD detection community.

UPDATE AFTER AUTHOR RESPONSE:

After reading the authors' responses to mine and other reviewers, I re-iterate my initial stance that this is a good paper which would be valuable to the community. I believe that focussed investigation on existing research trends, as carried out in this paper, is just as valuable as the proposals of novel methods.

I maintain my rating of '8, Good Paper'.

---

> ### Author Response · Authors · 2022-11-17
> **Response to Reviewer YM1z**
>
> We thank the reviewer for their feedback and encouraging remarks.
>
> We have updated the paper (changes marked in teal) to improve the clarity of notation as per your feedback, as well as to include a reference to recent work from Vaze et al.
>
> As for the "trivially verifiable or falsifiable" question, this is a valuable suggestion; we intend to include histograms of the gradient norms for all (ID,OOD) splits in the appendix.

---

### Official Review · Reviewer_ACRA · 2022-10-25

**Confidence:** 4
**Correctness:** 3
**Technical Novelty And Significance:** 3
**Empirical Novelty And Significance:** 3
**Recommendation:** 6

**Clarity, Quality, Novelty And Reproducibility:**

- The paper is written clearly and the contributions are clearly stated, and the results are interesting.


**Strength And Weaknesses:**


- The paper is written clearly and the contributions are clearly stated.
- Some conclusions are not fully supported/could use additional explanations
Questions/comments
- The results have high variance (cf. Fig 2), in this case the paper would greatly benefit from studying the performance of the proposed OOD methods on additional datasets. Conversely, are the authors proposed that using ||h||_{0.3} x energy should be the OOD score to use across different tasks/models?
- Eq 6 requires a more through explanation. Why does it imply learning a true mapping?
- Can the authors elaborate on their explanation why varsum and exgrad have subpar performance?
- The authors don't include AUPR results which should be included. While AUC-ROC shows us the tradeoff between true positive and false positive predictions, and the AUC-PR shows us the tradeoff between precision (low false positive) and recall (low false negative) predictions.
- Do the authors believe that the feature extraction / familiarity hypothesis [1] holds for the last layer?

[1] Dietterich and Guyer 2022

**Summary Of The Paper:**

This paper evaluates different methods for OOD detection which use and do not use gradient information. They decompose several methods as in [1] into two components related to the encoding and output probability vectors.
Through their decompositions they propose different OOD detection methods not based on gradients which outperform benchmarks such as grad norm [1] in some settings.

[1] Huang et al. 2021

**Summary Of The Review:**


The paper is very interesting and beautifully written. However, some statements and results could benefit from additional clarity. I would be glad to see this paper accepted if the authors can address all the reviewer’s questions (above) as it has an interesting take on ongoing OOD detection discussions in the literature.

---

> ### Author Response · Authors · 2022-11-16
> **Response to Reviewer ACRA**
>
> We thank the reviewer for their thoughtful feedback.
>
> Firstly, while $\lVert \mathbf{h}\rVert_{0.3} \times \text{energy}$ does seem to be a strong performer for the baselines we chose, we believe the takeaway here is that there are methods derived from the $UV$ framework have stronger performance than those derived from gradients. As such, we assert that research efforts should be focused on this $UV$ setting rather than gradients.
>
> Secondly, note that by “true mapping”, we mean a mapping that is not degenerate, where a “degenerate mapping” is defined as one that maps to the same value for all inputs. We have updated the paper to help clarify this detail (changes shown in teal). Note that every gradient-based OOD estimator is associated with a loss function, specifically the function defining the value whose gradient is being computed. We can interpret this loss function as defining an optimisation problem, and the gradient points in the direction of a local optimum for this optimisation problem. We note that the loss function inside the parentheses of equation 6 defines an optimisation problem that, for sufficiently expressive networks, cannot have degenerate mappings as its solution. This is because the 2nd term inside the parentheses penalizes choices of $\theta$ that parameterise a degenerate mapping. Note that this is in contrast to other loss functions, such as those used in ExGrad and GradNorm, where degenerate mappings are in the solution spaces of their associated optimisation problems. The motivation for the BatchGrad loss function is to design an experiment where gradients are guaranteed to point towards a true, non-degenerate mapping, giving a better chance of observing the Feature Extraction Hypothesis in action, should it be true. In particular, true mappings are required in order to “extract” features. BatchGrad encourages this prerequisite, and yet even here, we see that restricting gradients to the final layer results in improvements to performance. In other words, if the Feature Extraction Hypothesis were true, then we would be especially likely to see a reduction in OOD detection performance when restricting gradients to the final layer of BatchGrad, and yet we see the opposite.
>
> Lastly, given that representation results only apply to networks of at least depth 2 (e.g. universal approximation theorems), it is implausible that gradients at the final layer are attempting to extract rich features from test time data. Therefore we do not believe that the Feature Extraction Hypothesis explains the performance of gradient-based OOD methods restricted to the final layer weights. In addition, we have not provided experimental evidence to support or reject the Familiarity Hypothesis: amongst other results, [1] provided data from object-blurring experiments to isolate the contributions of on-object features to the logit scores and found that the reduction in logit scores for OOD data was largely attributable to the absence of familiar features. We have not performed experiments of this nature in this paper. However, the Familiarity Hypothesis does not contradict any of our results, and it is possible that the utility in $\lVert \mathbf{h}\rVert_{0.3}$ is, in effect, “counting” the number of familiar features present in a test image, similar to the thresholding experiments in [1].
>
> [1] Dietterich and Guyer 2022

---

### Comment · Area_Chair_pkCj · 2022-12-06
**Choice of architecture**

Dear authors,

During the discussion the reviewers and AC raised and emphasized the following issue: The choice of architecture (ResNetv2-101, what is the precise reference to this model? is it the 1x or 3x?) is not explained and seems odd. Are there any pretraining effects that influence the results and analysis? A more convincing demonstration would be if the same effects were shown using a mainstream contemporary architecture (i.e., EfficienNet, RegNet, MobileNet) and the effects of any pretraining were isolated. Thus, even Resent-50 without pretraining would be more convincing.

Sincerely,

AC

---

### Decision · Program_Chairs · 2023-01-20

**Decision:**

Reject

**Justification For Why Not Higher Score:**

N/A

**Justification For Why Not Lower Score:**

N/A

**Metareview: Summary, Strengths And Weaknesses:**

This paper evaluates a family of methods for OOD detection with and without gradient information. A decomposition of those methods as in [1] in terms of two components (related to the encoding and output probability vectors) is considered. Through this decompositions the paper examines the performance of a number of combination methods. The main finding is that methods not based on gradients outperform baselines such as gradnorm [1] in some settings.

While the paper proposes interesting analyses, several concerns remained after the rebuttal. An implementation choice that is not explained by the authors raises concerns on the generality of their main finding. Specifically, the choice of network (architecture and training regime) is not explained and since it involves heavy pretraining, it casts doubts on the generality of the observations. A more convincing demonstration would be if the same effects were shown using a mainstream contemporary architecture (i.e., EfficienNet, RegNet, MobileNet) and the effects of any pretraining were isolated. In this respect even Resent-50 without pretraining could have been more convincing. We encourage the authors to exhibit this behavior on a several networks, with and without pretraining.

[1] Huang et al. 2021

**Summary Of Ac-Reviewer Meeting:**

The AC and reviewers conducted a virtual meeting where the paper was throughly discusssed. Everybody expressed their views and the during the meeting the choice of network issue was raised.